# TAGA: Self-supervised Learning for Template-Free Gaussian Avatar

## Abstract

Decoupling from customized parametric templates marks an integral leap towards creating fully flexible, animatable avatars. In this work, we introduce TAGA (Template-free Animatable Gaussian Avatars), the first template-free, Gaussian-based solution for the reconstruction of animatable avatars from monocular videos, which offers distinct advantages in fast training and real-time rendering. Constructing template-free avatars is challenging due to the lack of predefined shapes and reliable skinning anchors to ensure consistent geometry and movement. TAGA addresses this by introducing a self-supervised method which guides both geometry and skinning learning leveraging the one-to-one correspondence between canonical and observation spaces. During the forward mapping phase, a voxel-based skinning field is introduced to learn smooth deformations that generalize to unseen poses. However, without template priors, forward mapping often captures spurious correlations of adjacent body parts, leading to unrealistic geometric artifacts in the canonical pose. To alleviate this, we define Gaussians with spurious correlations as "Ambiguous Gaussians" and then propose a new backward mapping strategy that integrates anomaly detection to identify and correct Ambiguous Gaussians. Compared to existing state-of-the-art template-free methods, TAGA achieves superior visual fidelity for novel views and poses, while being **60** × faster in training (**0.5** hours *vs* 30 hours) and **560** × faster in rendering (**140** FPS *vs* 0.25 FPS). Experiments on challenging datasets that possess limited pose diversity further demonstrate TAGA's robustness and generality. Code will be released.

## 1 Introduction

Parametric templates, such as SMPL [1] and SMAL [2], play a pivotal role in the field of 3D avatar reconstruction, providing two essential priors, including mesh vertices, which anchor the model's geometry with precise prior shape; and vertex skinning weights, which determine how each vertex moves relative to bone joints. However, creating these templates requires labor-intensive 3D scanning and manual annotation [1–6], which limits their application in various real-world object categories.

Recent advancements in template-free approaches have sought to address the limitations of traditional methods by utilizing 3D poses instead of predefined templates. Though much progress has been made, a fundamental challenge still remains: how to accurately recover the canonical model (Fig. 1(b)) from posed observations (Fig. 1(a)). To reverse the observations and construct a canonical body model, implicit template-free methods often rely on learning inverse skinning or complex iterative root-finding algorithms to establish canonical correspondences that fits the sample points in observation space. However, both approaches heavily rely on rich pose data as input, which can be impractical due to the high costs asso-

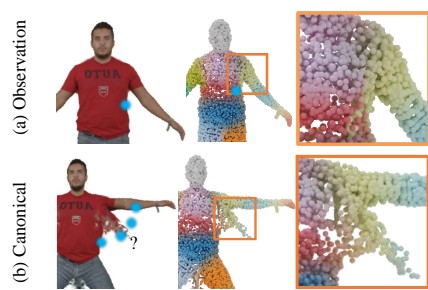

Figure 1: Canonical ambiguity: Gaussians render observed poses well (a), but produce significant artifacts in canonical space (b).

ciated with data collection and annotation. When pose data is sparse, recovering canonical models presents an ill-posed problem, as multiple canonical models could potentially fit the limited observations. Thus, although these methods may achieve reasonable reconstructions in the observation

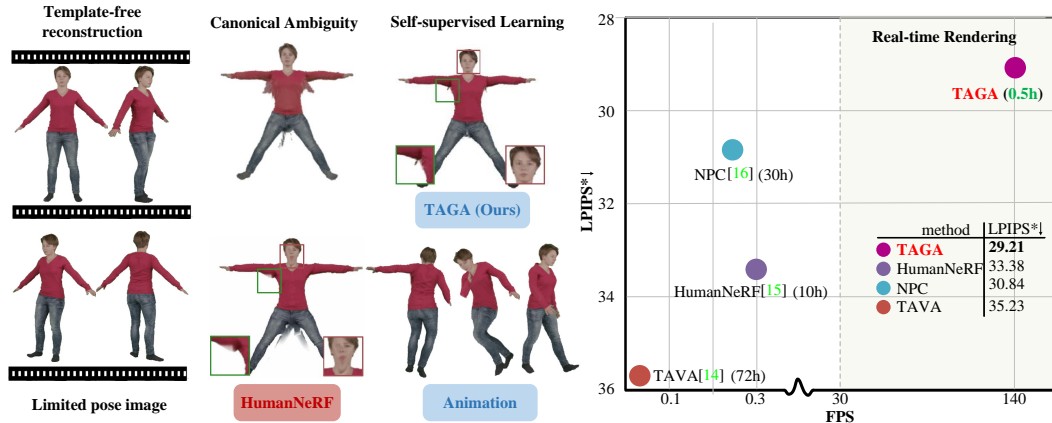

Figure 2: We propose TAGA, the first template-free Gaussian-based method that generates avatars in **30** minutes, with real-time rendering up to **140** FPS. Another key advantage of TAGA is its ability to handle low pose diversity from monocular video inputs. Without relying on templates, TAGA employs self-supervised learning to resolve ambiguities in the canonical space, resulting in realistic and animatable avatars.

space, they often face spurious correlations between adjacent parts and severe geometric artifacts in the canonical space (Fig. 1(b)). Furthermore, these methods typically focus only on reproducing a limited set of 2D observations, which prevents them from recognizing ambiguities in canonical reconstruction; unless sufficient data is provided, they cannot resolve these ambiguities and, as a result, cannot optimize an accurate canonical model.

In this work, TAGA utilizes 3D Gaussians as the canonical representation, which has been widely shown to provide accurate observation space reconstructions through forward mapping2. However, due to the flexibility of Gaussians, this canonical ambiguity is still pronounced in a template-free scenario, significantly hindering the ability to animate the avatar. To deal with this challenge, TAGA exploits the explicit one-to-one correspondence of 3D Gaussians, which, through skinning, maintains a bijective mapping between the canonical and observation spaces. By using this correspondence as an anchor, and given that the visible observations can be accurately reconstructed, it follows naturally that with the correct skinning, we can also achieve an accurate canonical reconstruction. Building on this insight, we develop a coarse-to-fine self-supervised framework. First, during forward mapping, we learn a voxel-based skinning field to obtain a suboptimal canonical reconstruction. Then, we progressively correct the "Ambiguous Gaussians" – those with incorrect skinning—in the observation space, fixing them point by point. These Ambiguous Gaussians arise from spurious correlations betweeen adjacent body parts, where the skinning does not align with the semantics of their positions. In the absence of skinning prior for part assignment, we employ an anomaly detection algorithm – specifically, a bone-based GMM – to mine spatial and semantic cues in the observation space, enabling us to identify and correct the ambiguous Gaussians in an unsupervised manner. The corrected Gaussians are then mapped back to refine the original canonical model.

Compared to traditional implicit representation approaches, TAGA focuses on iteratively refining the forward mapping via our proposed new backward mapping strategy, fully exploiting the speed advantage of 3D Gaussian splatting in forward rendering. This design overcomes the generalization limitations of inverse skinning and eliminates the computational overhead associated with root-finding. As a result, TAGA enables rapid reconstruction of an animatable avatar from monocular video in just **30** minutes, achieving real-time rendering at over **140+** FPS. To our knowledge, this performance exceeds that of any other template-free method. Our contributions are threefold:

- We present TAGA, the first Gaussian-based framework for building animatable 3D avatars without parametric templates, enabling many advantages such as high-quality reconstruction, efficient training, and real-time rendering.
- We leverage inherent one-to-one-correspondence of 3D Gaussian as an anchor to jointly refine canonical geometry and skinning in a self-supervised manner.
- We propose a new backward mapping apporach that integrates anomaly detection to handle canonical ambiguity, addressing spurious correlations in template-free avatar reconstruction.

| Method | Template Free | Backward Mapping | Explicit Representation | Monocular Input | Real-time Rendering | Training Time |
|---|---|---|---|---|---|---|
| 3DGS-Avatar [10] | ✗ | ✗ | ✓ | ✓ | ✓ | 30m |
| GART [11] | ✗ | ✗ | ✓ | ✓ | ✓ | 3m |
| InstantAvatar [12] | ✗ | ✓ | ✗ | ✓ | ✗ | 5m |
| InstantNVR [13] | ✗ | ✓ | ✗ | ✓ | ✗ | 5m |
| TAVA [14] | ✓ | ✗ | ✗ | ✗ | ✗ | 72h |
| HumanNeRF [15] | ✓ | ✓ | ✗ | ✓ | ✗ | 10h |
| NPC [16] | ✓ | ✗ | ✓ | ✓ | ✗ | 30h |
| **TAGA (Ours)** | ✓ | ✓ | ✓ | ✓ | ✓ | 0.5h |

Table 1: Differences between TAGA and existing representative methods.

We conduct extensive experiments on the widely-used monocular dataset ZJU-MoCap [7] (§4.2) and the established, single-pose-dominant dataset PeopleSnapshot [8] (§4.2). Compared to existing template-free competitors, TAGA achieves state-of-the-art reconstruction quality, improving LPIPS* by **1.6** over NPC on ZJU-MoCap and by **7.0** LPIPS* over HumanNeRF on PeopleSnapshot. In addition, we evaluate TAGA on canonical pose and challenging motion sequences from AIST++ [9], demonstrating its robustness in canonical reconstruction even under extreme single-pose scenarios. Ablation studies further confirm the effectiveness of our framework design (§4.3).

## 2 RELATED WORK

**Templates-free Reconstruction Methods.** To eliminate reliance on parametric templates, various methods focus on building template-free animatable avatars. One prominent direction [15, 17–24], exemplified by HumanNeRF [15], compensates for the lack of shape priors by learning inverse skinning to map observed poses to a canonical space, but struggles with generalization to new poses. Another mainstream approaches [14, 25–30], represented by TAVA [14] and ARAH [30], perform complex and time-consuming iterative root-finding algorithm to search for correct canonical correspondences of points in observation space. More recently, NPC [16] uses sparse feature point clouds as anchors to accelerate the backward mapping of query points. However, since each sampling point requires querying K-nearest anchors during both forward rendering and backward mapping, it fails to fully leverage the speed advantages of explicit representations. All the above methods incur significant computational overhead during the mapping process, as they require extensive querying to establish correspondences between observation space points and their canonical counterparts. As a result, both training and rendering speeds are significantly slowed.

The 3D Gaussian representation, which has been widely adopted in SMPL-based human models [11, 33–41], holds the potential to overcome the aforementioned limtations, with enhanced speed, superior quality, flexible topology, and natural one-to-one correspondence [31–33]. Despite these advantages, it is surprising that template-free approaches for animatable avatar reconstruction based on Gaussian representations remain unexplored. In this work, we extend 3D Gaussian splatting to template-free avatars, achieving state-of-the-art synthesis quality on both novel view synthesis and unseen pose synthesis with just minutes of training and real-time rendering at over **140+** FPS. Table 1 provides a comparison between TAGA and recent representative animatable avatar reconstruction methods.

**Template-free Canonical Appearance Modeling.** In template-free scenarios, the absence of parametric templates requires learning canonical body geometry from scratch. NPC [16] sidesteps the problem by extracting explicit point clouds from existing part-based body models. Despite that, the use of fixed point clouds lacks flexibility [42], making it difficult to handle complex deformations and hindering end-to-end learning. Other traditional methods, whether relying on root-finding [14, 27, 30] or inverse skinning [15, 43], require rich pose data as input. When pose diversity is limited, these methods tend to learn spurious correlations in self-contact regions.

TAGA uses self-supervised learning to reconstruct the geometry and skinning of animatable avatars from a limited set of videos and poses, without relying on predefined templates. A conceptually related approach is SCANimate [48], which weakly supervises the reconstruction of clothed human bodies from raw scans by enforcing consistency between forward and inverse skinning. However, SCANimate requires learning inverse skinning and depends on skinning from SMPL template to supervise both forward and inverse skinning. In contrast, TAGA learns only forward skinning and

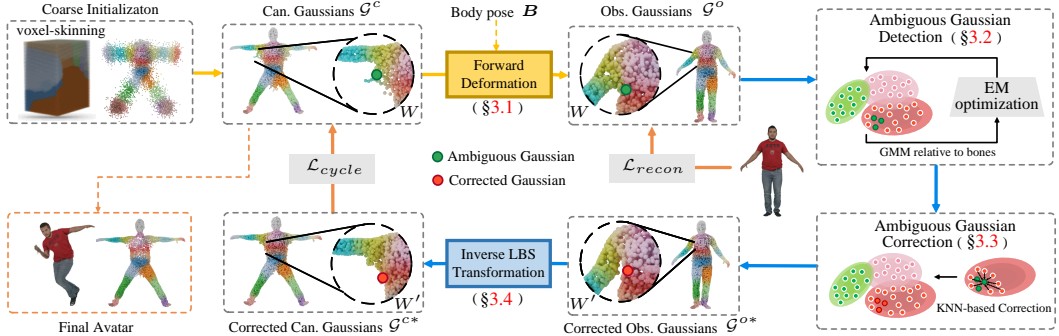

Figure 3: Overall framework: Given a pose, TAGA transforms canonical 3D Gaussians to the observation space through the Forward Deformation Module. To resolve ambiguities between adjacent body parts, TAGA detects and corrects Ambiguous Gaussians in observation space. Finally, TAGA maps these corrected Gaussians back to the canonical space using the inverse LBS transformation to guide the original canonical Gaussians.

introduces a new backward mapping strategy specifically designed for explicit Gaussians. Thanks to the explicit nature of Gaussians, our backward mapping strategy fundamentally differs from the Nerf-based counterparts. Rather than focusing on establishing dense correspondences to achieve a reconstruction that merely fits the input data, we take it a step further by using these correspondences as anchors. This allows us to transfer spatial and semantic information from the observation space back to the canonical space, thereby resolving ambiguities and improving the overall reconstruction.

## 3 METHODOLOGY

**Overview.** Given monocular videos and their corresponding poses, TAGA jointly learns the geometry and skinning field of an animatable avatar, without relying on parametric templates. A overview of pipeline is shown in Fig. 3. During forward mapping, a voxel-based skinning field is learned to deform Gaussian representations from the canonical space to the observation space (§3.1). To tackle spurious correlations between adjacent body parts in the absence of template priors, we detect Ambiguous Gaussians which are affected by spurious correlations. This is achieved through a bone-based GMM (§3.2), which enalbes us to correct the skinning of these Gaussians for proper alignment with body semantics (§3.3). The corrected Gaussians are then remapped back to the canonical space, with soft constraints guiding the canonical geometry and skinning field (§3.4). In addition, implementation details are provided in Appendix §C.

### 3.1 FORWARD DEFORMATION

**Canonical Gaussian Representation.** TAGA uses Gaussians $\mathcal{G}$ as the basic representation, defining them in the canonical space to model the avatar's appearance and shape. Each 3D Gaussian $g \in \mathcal{G}$ is characterized by its position $\boldsymbol{p}$, covariance $\boldsymbol{\Sigma}$, opacity $\alpha$, spherical harmonics coefficients $\phi$, rotation $\boldsymbol{R}$ and $\boldsymbol{S}$. In this study, the term "template-free" refers to the exclusion of mesh vertices and skinning annotations typically provided from parametric templates. Instead, we initialize the 3D Gaussians by sampling points from a Gaussian distribution centered at the midpoints of each bone, with the distribution's standard deviation empirically adjusted based on the head/torso and distal joints.

**Voxel-based Skinning Field.** In template-free scenarios, the lack of skinning supervision from parametric templates and geometric priors hampers the direct application of point-to-point supervision on the skinning weights of Gaussians. A remedy is to sample points from bones and impose rigid constraints on their skinning. However, traditional MLPs struggle to effectively utilize the limited supervision provided by these sampled bone points, often overfitting to the few points rather than generalizing across the entire 3D space. Additionally, the dynamically changing Gaussians during training exacerbate this challenge. To address these obstacles, we employ a low-resolution fixed voxel grid to distill the skinning weight field from the MLP, where the MLP predicts skinning weights only on the grid. The skinning weights for each 3D Gaussian $\boldsymbol{p}^c$ in the canonical space are then queried through trilinear interpolation from the voxel grid:

$$\boldsymbol{W} = \texttt{interp}(\texttt{MLP}(\boldsymbol{V}), \boldsymbol{p}^c) \in \mathbb{R}^{N \times K}, \tag{1}$$

where `interp` refers to the trilinear interpolation operation, $N$ denotes the number of Gaussians and $K$ reprensents the number of bones. The voxel-based skinning field presents key advantages that enhance its effectiveness. First, the fixed voxel grid stabilizes training by limiting the MLP to predefined points, avoiding the influence of variations in Gaussian positions and numbers. Experiments show that a resolution of just $64 \times 64 \times 16$ suffices for accurate skinning reconstruction (Table S2, §E). Second, it enables effective regularization, as skinning constraints can smoothly propagate from bones to nearby areas, providing a solid initialization. Lastly, the integration of linear interpolation with MLP enhances smoothness, improving generalization to new poses.

**Linear Blend Skinning (LBS) Transformation.** With the skinning weights $\boldsymbol{W}$, the canonical Gaussians are transformed to the observation space using LBS transformation matrix $\boldsymbol{T}$, defined as:

$$\boldsymbol{T} = \sum_{k=1}^{K} \boldsymbol{W}_k \boldsymbol{B}_k \in \mathbb{R}^{N \times 4 \times 4}, \tag{2}$$

where $\boldsymbol{B} = [\boldsymbol{B}_1, \ldots, \boldsymbol{B}_K] \in \mathbb{R}^{K \times 4 \times 4}$ denotes the bone transformations. To accurately reposition $\boldsymbol{p}^c$ and reorient $\boldsymbol{R}^c$ into the observation space based on the input pose, we apply the full transformation matrix $\boldsymbol{T}$ to the position, and the upper-left $3 \times 3$ submatrix $\boldsymbol{T}_{1:3,1:3}$ to the rotation, as follows:

$$\boldsymbol{p}^o = \text{LBS}(\boldsymbol{W}, \boldsymbol{B}, \boldsymbol{p}^c) = \boldsymbol{T}\boldsymbol{p}^c, \qquad \boldsymbol{R}^o = \text{LBS}_{1:3,1:3}(\boldsymbol{W}, \boldsymbol{B}, \boldsymbol{R}^c) = \boldsymbol{T}_{1:3,1:3}\boldsymbol{R}^c. \tag{3}$$

**Rendering by Gaussian Splatting.** Once the canonical Gaussians are transformed to the observation space, we render the image using the efficent differentiable rasterizer from 3D-GS [52].

## 3.2 Ambiguous Gaussian Detection

This module aims to accurately identify Ambiguous Gaussians – whose skinning weights do not align with their expected skinning. Typically, the skinning of Gaussians is primarily influenced by their spatial relationship with the skeletons [53–57]. A rough estimation of skinning weights can be achieved by constructing a bone-based GMM. Each bone is associated with a Gaussian distribution that define its region of influence in 3D space. The skinning weights are then estimated as the likelihood that a Gaussian in 3D space is influenced by the GMM component centered on a particular bone, providing a rough yet effective approximation.

**GMM for Skinning**. For each bone $j$, a GMM component is defined centered at the bone's midpoint. The bone's orientation determines one axis of the Gaussian ellipsoid, with two orthogonal axes completing the basis. Semi-axis lengths are estimated using points with skinning weights above $\tau = 0.2$, taking the 85th percentile of their projected distances onto each axis. The skinning weight of $j$ -th bone for the $i$-th Gaussian position $\boldsymbol{p}_i^o$ in the observation space is estimated as follows:

$$\hat{\boldsymbol{W}}_{ij} = p(\boldsymbol{p}_i^o | j) = \frac{\mathcal{F}(\boldsymbol{p}_i^o; \boldsymbol{\mu}_j, \boldsymbol{\Sigma}_j)}{\sum_{k=1}^{K} \mathcal{F}(\boldsymbol{p}_i^o; \boldsymbol{\mu}_k, \boldsymbol{\Sigma}_k)} \in [0, 1], \tag{4}$$

where $\mathcal{F}(\boldsymbol{p}_i^o; \boldsymbol{\mu}_j, \boldsymbol{\Sigma}_j)$ is the probability density of $\boldsymbol{p}_i^o$ with respect to the $j$-th GMM component.

**Ambiguous Gaussian Definition.** Ambiguous Gaussians are detected by comparing the GMM-estimated skinning weight $\hat{\boldsymbol{W}}$ with the current skinning weight $\boldsymbol{W}$. For each Gaussian $g_i$, a confidence score $\boldsymbol{S}$ is computed using the Jensen-Shannon divergence (`JSD`):

$$\boldsymbol{S} = 1 - \text{JSD}(\hat{\boldsymbol{W}}_i \| \boldsymbol{W}_i) \in [0, 1]. \tag{5}$$

Gaussians with $S_i \le \alpha$ are classified as *Ambiguous Gaussians*, indicating a significant deviation from expected skinning weights. The set of Ambiguous Gaussians is denoted as $\mathcal{A} = \{g_i \mid S_i \le \alpha\}$, while the rest are classified as normal, denoted as $\overline{\mathcal{A}}$. All detected Ambiguous Gaussians will subsequently receive new skinning weights through the correction module.

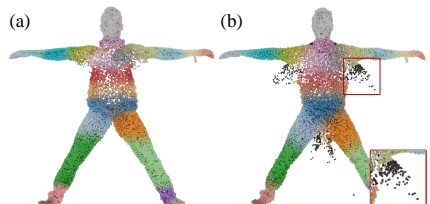

Figure 4: Illustrations of (a) GMM-estimated skinning weights and (b) detected Ambiguous Gaussians (marked as black points).

**GMM Parameter Optimization and Ambiguous Gaussian Detection.** To better detect Ambiguous Gaussians, we iteratively apply the Expectation-Maximization (EM) algorithm during each backward step to optimize the GMM parameters. The GMM parameters from the final iteration are used to identify ambiguous Gaussians, serving as the detection result for this backward step.

- $E$**–Step.** Estimate skinning weights $\hat{\boldsymbol{W}}$ using current GMM parameters and compute confidence scores $S$ to identify Ambiguous Gaussians $\mathcal{A}$ and normal Gaussians $\overline{\mathcal{A}}$.
- $M$**–Step.** Update the semi-axis lengths of GMM components using only the normal Gaussians $\overline{\mathcal{A}}$ detected in the last $E$-step.

### 3.3 Ambiguous Gaussian Correction

Given the detected Ambiguous Gaussians, this module aims to assign more appropriate skinning weights to them. To achieve this, we propose using the KNN algorithm to select the skinning weights of the $K$-Nearest normal Gaussians around each Ambiguous Gaussian. We then compare these weights with the estimated skinning weight of the current Ambiguous Gaussian and choose the one with the highest confidence. Specifically, for each Ambiguous Gaussian $g_i$, let $\mathcal{N}(i)$ denote its $K$-Nearest normal neighbors. We assign a new skinning weight to $g_i$ as follows:

$$\boldsymbol{W}'_i = \boldsymbol{W}_{n^*}, \quad \text{where} \quad n^* = \arg \max_{n \in \mathcal{N}(i)} (1 - \text{JSD}(\hat{\boldsymbol{W}}_i \parallel \boldsymbol{W}_n)). \tag{6}$$

### 3.4 Inverse LBS Transformation

For implicit representation, there is no direct correspondence between points in the canonical and observation space, and it is difficult to ensure bijectivity [58–60]. Classical backward mapping strategy primarily sought to establish correspondences from observation space ($\boldsymbol{x}^o$) to canonical space ($\boldsymbol{x}^c$), often expressed mathematically as solving for $\boldsymbol{x}^c$ where $\text{LBS}(w(\boldsymbol{x}^c), \boldsymbol{x}^c, \boldsymbol{B}) = \boldsymbol{x}^o$. Since implicit representations do not explicitly store points, we denote positions in 3D space with $\boldsymbol{x}$, while $w$ establishes a continuous skinning field in the canonical 3D space. As the relationship between $\boldsymbol{x}^o$ and $\boldsymbol{x}^c$ remains unknown, it is impossible to obtain an analytical solution for $\boldsymbol{x}^c$ directly [61, 62]. Therefore, existing methods resort to cumbersome and time-consuming iterative root-finding algorithms, which often require tens of hours of training.

In contrast, the explicit nature of Gaussians gives a one-to-one correspondence [63, 64] between the canonical Gaussians $\mathcal{G}^c$ and those $\mathcal{G}^o$ in the observation space. The skinning weights $\boldsymbol{W}$ in the canonical space are directly associated with the Gaussians themselves. The Gaussians act as anchors for transferring the skinning weights from the canonical space to the observation space. Thus, according to Eq. 3, $\boldsymbol{p}^c$ can be elegantly obtained as $\boldsymbol{T}^{-1}\boldsymbol{p}^o$. Given the estimated weights $\boldsymbol{W}'$ derived from observation space, the positions $\boldsymbol{p}^{c*}$ of the corrected canonical Gaussians are computed by applying an inverse LBS transformation using the adjusted skinning weights $\boldsymbol{W}'$ as follows:

$$\boldsymbol{p}^{c*} = \left(\sum\nolimits_{k=1}^{K} \boldsymbol{W}'_k \boldsymbol{B}_k\right)^{-1} \boldsymbol{p}^o. \tag{7}$$

**Cycle Consistency Loss.** Cycle consistency loss $\mathcal{L}_{cycle}$ is built on the hypothesis that, if Ambiguous Gaussians are correctly identified and corrected, their mapping back to the canonical space will perfectly recover the canonical model. Unfortunately, since the detection and correction process is conducted in an unsupervised manner, the mapped canonical Gaussians $\mathcal{G}^{c*}$ cannot be used as new canonical Gaussians directly. Instead, we use them as soft constraints to guide the refinement of the original canonical Gaussians $\mathcal{G}^c$. The cycle consistency loss $\mathcal{L}_{cycle}$ is composed of a geometry consistency loss ($\mathcal{L}_{geo}$) and a skinning consistency loss ($\mathcal{L}_{skin}$). Note that cycle consistency loss $\mathcal{L}_{cycle}$ is applied only to the detected Ambiguous Gaussians.

- **Geometry Consistency Loss ($\mathcal{L}_{geo}$):** This loss encourages the positions of the original canonical Gaussians $\boldsymbol{p}$ align with the corrected canonical Gaussians $\boldsymbol{p}^*$, enhancing the geometric consistency of the model. The loss is defined as:

$$\mathcal{L}_{geo} = \frac{1}{|\mathcal{A}|} \sum_{g_i \in \mathcal{A}} \|\boldsymbol{p} - \boldsymbol{p}^*\|_1 \,,$$

- **Skinning Consistency Loss ($\mathcal{L}_{skin}$):** This loss refines the skinning field by ensuring that the skinning weights at the positions $\boldsymbol{p}^*$ of the corrected canonical Gaussians $\mathcal{G}^{c*}$ align with the corrected skinning weights $\boldsymbol{W}'$. The loss is given by:

$$\mathcal{L}_{skin} = \frac{1}{|\mathcal{A}|} \sum_{g_i \in \mathcal{A}} \|w(\boldsymbol{p}^*) - w'(\boldsymbol{p}^*)\|_2^2 \,,$$

where $w(\boldsymbol{p^*})$ denotes the original skinning weights at position $\boldsymbol{p}$, and $w'(\boldsymbol{p^*})$ represents the corrected skinning weights of the detected Ambiguous Gaussians.

**Remark:** Our backward mapping strategy boasts several attractive qualities. ❶ **Transparency**: Central to our backward mapping strategy is the detection of Ambiguous Gaussians – a process that is straightforward, intuitive, and readily interpretable by humans. Unlike traditional implicit methods that operate as black boxes, our approach ensures full transparency in both detection and correction stages. Whether identifying or correcting Ambiguous Gaussians, or dealing with the resulting canonical Gaussians, each intermediate steps can be viewed and inspected (See Fig. 4). ❷ **Flexibility**: Our anomaly detection framework is not tied to a specific algorithm. Since we focus solely on using anomaly detection algorithms to perform unsupervised detection of Ambiguous Gaussians, TAGA can seamlessly integrate other point cloud anomaly detection methods into the current framework. ❸ **Robustness**: By integrating anomaly detection to capture overlooked spatial and semantic information, TAGA enables template-free reconstruction from limited pose data while resolving ambiguities and spurious correlations that typically arise from this ill-posed problem in the canonical space. ❹ **Efficiency**: TAGA utilizes the one-to-one correspondence of Gaussian representations to efficiently refine the canonical space, avoiding unnecessary exploration and focusing on incremental improvements from a suboptimal reconstruction.

### 3.5 TRAINING OBJECTIVE

**Bone Regularization Loss:** To encourage accurate skinning without parametric templates, we impose a rigid constraint by enforcing one-hot skinning weights at sampled points along each bone. The loss function is defined as: $\mathcal{L}_{bone} = \|\boldsymbol{W}_{sample} - \boldsymbol{W}_{gt}\|_2^2$, where $\boldsymbol{W}_{sample}$ represents the predicted skinning weights at the sampled points, and $\boldsymbol{W}_{gt}$ denotes the ground truth one-hot vectors.

**Loss Function.** The complete loss function includes the bone regularization loss $\mathcal{L}bone$, the cycle consistency loss $\mathcal{L}cycle$, and the reconstruction loss $\mathcal{L}_{recon}$. The full loss function is expressed as:

$$\mathcal{L} = \mathcal{L}_{recon} + \lambda_{bone}\mathcal{L}_{bone} + \mathcal{L}_{cycle}. \tag{8}$$

For detailed definitions and corresponding weights, please refer to the Appendix B.

## 4 EXPERIMENT

### 4.1 EXPERIMENTAL SETUP

**Datasets.** We evaluate our method using two established benchmarks:

- **ZJU-MoCap** [7] is a comprehensive dataset that captures a diverse range of human poses. For our experiments, we employ the monocular setup from InstantNVR[13], utilizing images from "camera 4" are used for training, while the remaining 22 cameras serve for evaluation. Our experiments are conducted on six specific subjects: 377, 386, 387, 392, 393, and 394.
- **PeopleSnapshot** [8] offers monocular videos of human subjects performing limited rotations in an A-pose. We follow the InstantAvatar[12] setup and conduct experiments on four sequences.

**Evaluation Metrics.** Following the widely adopted protocols [65], we evaluate novel view and pose synthesis using PSNR, SSIM, and LPIPS (scaled by 1000 for clarity).

**Competitors.** We compare TAGA with recent SOTA template-free and template-based methods. For ZJU-MoCap [7], we compare TAGA with template-free methods (TAVA [14], HumanNeRF [15], NPC [16]), as well as template-based methods, including NeRF-based methods (InstantAvatar [12], InstantNVR [13]) and Gaussian-based method GART [11]. For PeopleSnapshot [8], which features limited pose variations (self-rotating), we conduct experiments with the representative template-free method HumanNeRF and template-based methods (InstantAvatar and Anim-NeRF [65]).

**Reproducibility.** TAGA is trained on one RTX 3090 Ti GPU. Testing is conducted on the same machine. To guarantee reproducibility, our code and model weights will be released.

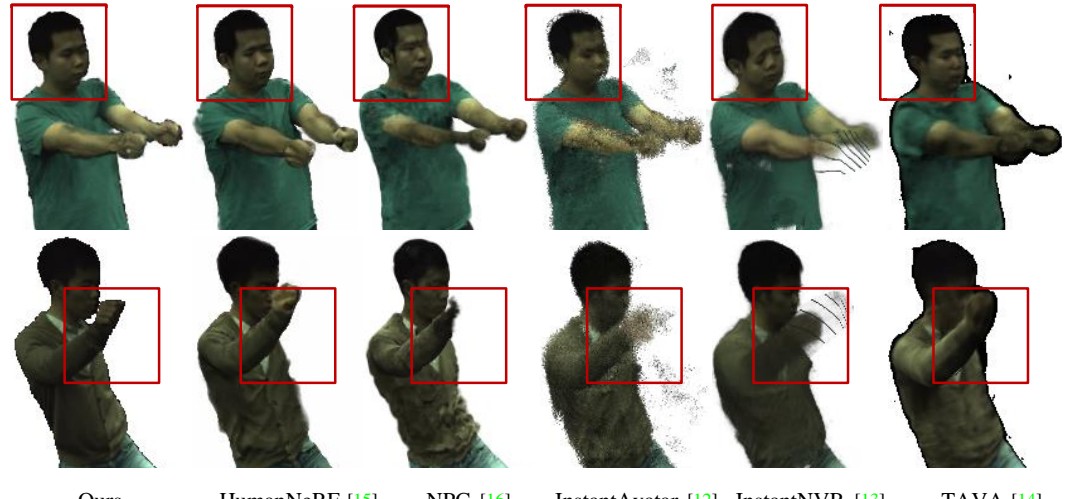

| Ours | HumanNeRF [15] | NPC [16] | InstantAvatar [12] | InstantNVR [13] | TAVA [14] |

Figure 5: Qualitative comparison on ZJU-MoCap [7] (§4.2).

## 4.2 COMPARISON RESULT

**Comparisons on ZJU-Mocap** [7]. As described in Table 2, TAGA provides notable performances over all template-free methods across all metrics, including PSNR, SSIM, and LPIPS, as well as state-of-the-art SMPL-based NeRF methods like InstantAvatar [12] and InstantNVR [13].

Compared to the SOTA template-free competitor NPC [16], TAGA exhibits substantial gains, achieving a PSNR of **31.22** compared to 30.76, an increase in SSIM from 0.969 to **0.977**, and a notable reduction in LPIPS* from 30.84 to **29.21**. Benefiting from the efficient 3D Gaussian splatting, TAGA reduces training time to **0.5** hours, which is **20** × faster than HumanNeRF [15] (10 hours) and **60** × faster than NPC (30 hours). In terms

Table 2: Quantitative results on ZJU-MoCap [7] (§4.2).

| Method | SMPL | GPU↓ | FPS ↑ | Novel view PSNR↑ | SSIM↑ | LPIPS*↓ |
|---|---|---|---|---|---|---|
| GART [11] | ✓ | 0.1h | 46.2 | 32.31 | 0.982 | 24.91 |
| InstantAvatar [12] | ✓ | 3m | 4.15 | 29.73 | 0.938 | 68.41 |
| InstantNVR [13] | ✓ | 5m | 2.20 | 31.01 | 0.971 | 38.45 |
| TAVA [14] | | 72h | 0.01 | 30.24 | 0.969 | 35.23 |
| HumanNeRF [15] | | 10h | 0.30 | 30.66 | 0.969 | 33.38 |
| NPC [16] | | 30h | 0.25 | 30.76 | 0.960 | 30.84 |
| **TAGA (Ours)** | | 0.5h | **140** | **31.22** | **0.977** | **29.21** |

of inference, TAGA achieves real-time rendering rates at **140** FPS, surpassing the implicit representation counterpart HumanNeRF (0.3 FPS) by **470** × and being **560** × faster than the explicit point cloud method NPC (0.25 FPS). Moreover, TAGA achieves comparable performance with the latest template-based Gaussian method, GART [11], without reliance on any template prior.

Qualitative comparisons for novel view synthesis are shown in Fig. 5. Methods like TAVA [14], InstantNVR [13], and InstantAvatar [12] employ traditional iterative root-finding algorithms for modeling canonical appearance. However, these methods face challenges in capturing high-frequency details like loose clothing, resulting in blurry outputs and occasional severe distortions. Human-NeRF [15] performs well overall, preserving details of loose clothing, but encounters difficulties with facial and hand details and shows artifacts along edges. The explicit method NPC [16] suffers from significant artifacts with loose clothing due to its reliance on fixed point clouds, which are unable to adapt to complex non-rigid deformations. In contrast, TAGA excels at reconstructing realistic high-frequency details like facial features, clothing, and hands, with fewer artifacts.

**Comparisons on PeopleSnapshot** [8]. For PeopleSnapshot, characterized by highly repetitive poses, TAGA demonstrates substantial improvement over the template-free baseline, HumanNeRF [15], which performs well on the ZJU-Mocap dataset. Quantitative results can be found in Table 3. As an example, in the male-3-casual sequence, notable enhancements are observed in PSNR (**29.12** *vs* 26.13), SSIM (**0.970** *vs* 0.955), and LPIPS* (**21.7** *vs* 27.7). Futhermore, TAGA significantly outperforms the SMPL-based method Anim-NeRF [65], while achieving performance on par with the leading SMPL-based method, InstantAvatar [12].

Table 3: Quantitative results on PeopleSnapshot [8] (§4.2).

| Method | GPU↓ | FPS↑ | female3-casual | | | female4-casual | | | male3-casual | | | male4-casual | | |
|---|---|---|---|---|---|---|---|---|---|---|---|---|---|---|
| | | | PSNR↑ | SSIM↑ | LPIPS*↓ | PSNR↑ | SSIM↑ | LPIPS*↓ | PSNR↑ | SSIM↑ | LPIPS*↓ | PSNR↑ | SSIM↑ | LPIPS*↓ |
| Anim-NeRF [65] | 13h | 0.1 | 23.87 | 0.950 | 34.6 | 24.37 | 0.945 | 38.2 | 24.94 | 0.943 | 32.6 | 24.71 | 0.947 | 42.3 |
| InstantAvatar [12] | 5m | 15 | 27.90 | 0.972 | 24.9 | 28.92 | 0.969 | 18.0 | 29.65 | 0.973 | 19.2 | 27.97 | 0.965 | 34.6 |
| HumanNeRF [15] | 5h | 0.1 | 23.82 | 0.948 | 37.6 | 26.76 | 0.960 | 23.7 | 26.13 | 0.955 | 27.7 | 24.46 | 0.936 | 50.2 |
| TAGA (Ours) | 30m | 140 | 24.99 | 0.956 | 32.2 | 28.40 | 0.967 | 20.8 | 29.12 | 0.970 | 21.7 | 26.73 | 0.958 | 36.5 |

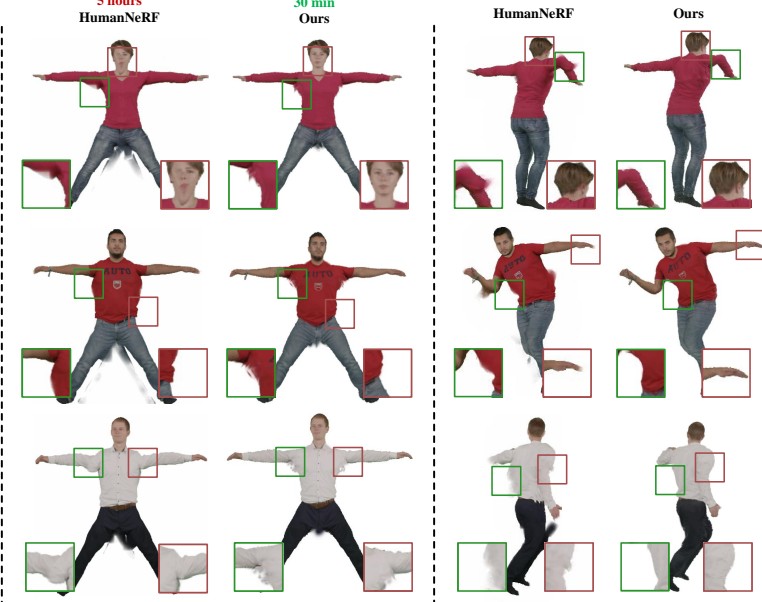

Figure 6: Qualitative results on PeopleSnapshot [8] (§4.2). We display reconstructed avatars from various viewpoints, canonical poses, and novel pose animations.

Fig. 6 presents the qualitative results of HumanNeRF [15] and TAGA for test views and novel poses. In test views, HumanNeRF faces difficulties in head reconstruction, leading to distorted facial details. This challenge stems from the ambiguous correspondences introduced by inverse skinning when attempting to reversing multiple observation. In contrast, TAGA benefits from the inherent one-to-one correspondence of explicit Gaussians, resulting in more consistent canonical reconstructions.

To further evaluate the animation capabilities of HumanNeRF [15] and TAGA, we animate models trained on PeopleSnapshot using canonical poses and challenging motion sequences from AIST++ [9]. As shown in Fig. 6, HumanNeRF performs poorly in canonical poses, with clear artifacts at the seam between the legs and an unrealistic reconstruction of the underarm geometry. This suggests that HumanNeRF struggles to resolve spurious correlation between body parts in close proximity. In contrast, TAGA successfully reconstructs accurate geometry, even without ground-truth annotation of canonical pose. Although minor noise is present, this is likely due to the occlusion of underarms and the region between the legs in the PeopleSnapshot dataset. Additionally, TAGA demonstrates significantly better generalization to novel poses, whereas HumanNeRF exhibits prominent artifacts around the clothing and joint boundaries.

## 4.3 DIAGNOSTIC EXPERIMENT

As the motions in ZJU-MoCap [7] and PeopleSnapshot [8] are usually repetitive, they lack ground truth annotations for uncommon poses, such as canonical pose. To evaluate the model's ability to animate out-of-distribution poses, we utilize SOTA SMPL-based method GART [11], to generate pseudo-ground truth for a set of representative poses sampled from the AIST++ [9]. Specifically, we use male-3-casual sequence from PeopleSnapshot to conduct our ablation experiments. The qualitative and quantitative results are shown in Fig. 7 and Table 4.

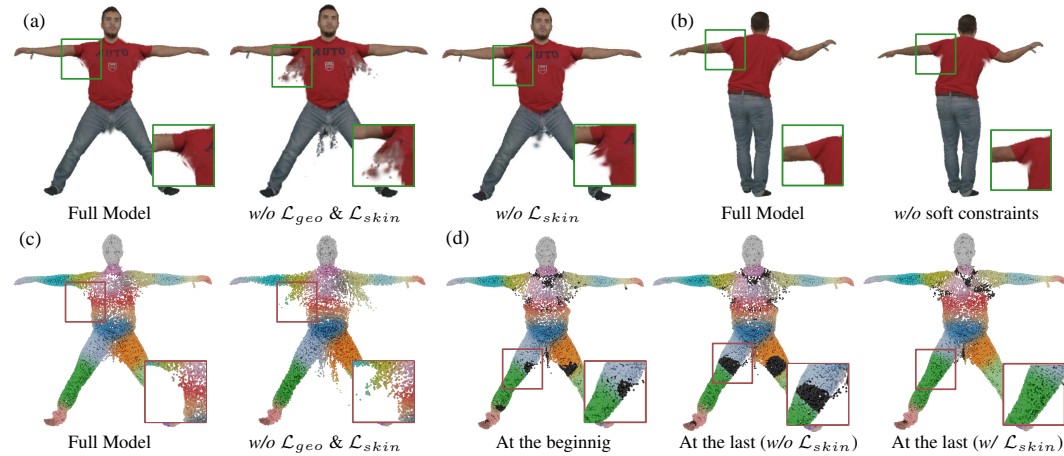

Figure 7: Diagnostic experiment (§4.3). (a) The effect of $\mathcal{L}_{geo}$ and $\mathcal{L}_{skin}$ on canonical appearance. (b) Artifacts *w/o* soft constraints. (c) Skinned point clouds for the full model and *w/o* $\mathcal{L}_{geo}$&$\mathcal{L}_{skin}$. (d) The impact of $\mathcal{L}_{skin}$ on identifying ambiguous Gaussians (marked as black points).

**Backward Strategy**. The quantitative results in Table 4 show a significant performance drop when different backward mapping strategies, such as $\mathcal{L}_{geo}$, $\mathcal{L}_{skin}$, and soft constraints, are removed. For instance, without $\mathcal{L}_{geo}$ and $\mathcal{L}_{skin}$, PSNR decreases to 24.10, SSIM drops 0.9393, and LPIPS* increases to 48.6, compared to the full model's PSNR of **28.89**, SSIM of **0.9685**, and LPIPS* of **23.1**. Similarly, removing $\mathcal{L}_{skin}$ leads to a performance drop (PSNR: 26.92, SSIM: 0.9567, LPIPS*: 32.1). Furthermore, the model without soft constraints also shows degradation across all metrics, indicating a decline in animation performance.

For qualitative results, Fig. 7(a) highlights noticeable artifacts at the joint seams, such as those between the arms and torso and between the legs. For example, in the armpit region, artifacts suggest that certain Gaussians should belong to the torso. However, Fig. 7(c) shows they are incorrectly influenced by the arm. Without backward mapping, these ambiguous Gaussians remain undetected and uncorrected, leading to severe artifacts in canonical space. As shown

Table 4: Ablative experiments on backward strategy for male-3-casual sequence (§4.3).

| Strategy | Novel pose | | |
|---|---|---|---|
| | PSNR↑ | SSIM↑ | LPIPS*↓ |
| *w/o* $\mathcal{L}_{geo}$ and $\mathcal{L}_{skin}$ | 24.10 | 0.9393 | 48.6 |
| *w/o* $\mathcal{L}_{skin}$ | 25.92 | 0.9567 | 32.1 |
| *w/o* soft constraints | 26.92 | 0.9567 | 32.1 |
| TAGA (Ours) | **28.89** | **0.9685** | **23.1** |

in Fig. 7(d), removing $\mathcal{L}_{skin}$ causes certain ambiguous Gaussians to be detected but not corrected throughout the entire backward phase. This occurs because $\mathcal{L}_{cycle}$ can optimize their positions in normalized space but cannot adjust the canonical skinning field. As a result, even though these Gaussians are correctly positioned, they still appear ambiguous due to the incorrect skinning field.

## 5 CONCLUSION

In this study, we tackle the challenge of reconstructing a canonical avatar from monocular videos with limited poses, without relying on parametric templates. We demonstrate that leveraging semantic and spatial cues from observations can compensate for the limited visual information during canonical reconstruction. Following this insight, we utilize the inherent bijectivity of Gaussians to design a coarse-to-fine forward-backward framework named TAGA that self-supervises the optimization of skinning and geometry in the canonical space. To this end, we propose a new backward mapping strategy that integrates anomaly detection to capture robust spatial and semantic inductive biases from the observed space, allowing for transparent correction of erroneous geometric artifacts caused by Ambiguous Gaussians in the canonical space. Extensive experiments demonstrate the robustness and efficiency of TAGA. We believe our contributions provide novel insights into template-free reconstruction, taking an important step towards overcoming the limitations imposed by parametric templates and observations with low diversity.

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

# SUMMARY OF THE APPENDIX

This appendix contains additional details for the ICLR 2025 submission, titled *TAGA: Template-Free Animatable Gaussian Avatars With Forward-Backward Consistency*. The appendix is organized as follows:

- §A provides the pseudo code of TAGA.
- §B introduces the details of loss function.
- §C introduces the training and inference details of TAGA.
- §D introduces the implementation details of baselines to compare.
- §E reports additional diagnostic experiments.
- §F gathers additional qualitative results on several dataset.
- §G discusses our limitations and directions of future work.

**Algorithm S1** Pseudo code for Ambiguous Gaussian Detection and Correction in a PyTorch-like style.

```
# N: number of Gaussians
# K: number of bones
# D_i: indicator function for Ambiguous Gaussians
# gamma: responsibilities in EM algorithm (N, K)
# pi_j: mixture weights in GMM (K, 1)
# ambiguous_gaussians: binary mask indicating Ambiguous Gaussians (N, 1)

def detect_ambiguous_gaussians(G, W, threshold):
    # Step 1: Initialize GMM parameters
    Sigma_j = initialize_covariance(G) # (K, 3, 3)
    pi_j = initialize_mixture_weights(G) # (K, 1)

    # Step 2: Perform EM algorithm
    for iteration in range(max_iterations):
        # E-step: Calculate responsibilities
        gamma = compute_responsibilities(G, Sigma_j, pi_j) # (N,K)
        gamma_prime = refine_responsibilities(gamma, W) # (N,K)

        # M-step: Update GMM parameters based on refined
            responsibilities
        Sigma_j = update_covariance_matrix(gamma_prime, G) # (K, 3, 3)
        pi_j = update_mixture_weights(gamma_prime) # (K, 1)

        # Step 3: Detect Ambiguous Gaussians using confidence scores
        S = compute_confidence_scores(gamma_prime, W) # (N, 1)
        ambiguous_gaussians = detect_ambiguous(S, threshold) # Binary
            mask indicating Ambiguous Gaussians (N, 1)

        if convergence_reached(ambiguous_gaussians):
        break

    # Output: Binary mask (0 or 1) indicating Ambiguous Gaussians
    return ambiguous_gaussians # (N, 1)

def correct_ambiguous_gaussians(G, W, ambiguous_gaussians):
    corrected_weights = W.clone() # (N, K)

    for g_i in range(len(G)):
        if ambiguous_gaussians[g_i] == 1: # Check if the Gaussian is
            ambiguous
        # Find K-nearest neighbors from non-Ambiguous Gaussians
        nearest_neighbors = find_K_nearest_neighbors(G[g_i], G[
            ambiguous_gaussians == 0])

        # Select the neighbor with the highest similarity score
        best_neighbor = select_best_neighbor(G[g_i], nearest_neighbors)

        # Update the weight for the Ambiguous Gaussian using the best
            neighbor's weight
        corrected_weights[g_i] = best_neighbor.W

    return corrected_weights # (N, K)
```

## A    PSEUDO CODE OF DNC AND CODE RELEASE

To facilitate a comprehensive understanding of TAGA, we provide pseudo code for our Ambiguous Gaussian Detection and Correction module in Algorithm S1.

## B    DETAILS OF LOSS FUNCTION

Our full loss function can be formulated as follows:

$$\mathcal{L} = \mathcal{L}_{recon} + \lambda_{bone}\mathcal{L}_{bone} + \mathcal{L}_{cycle}. \tag{9}$$

**Reconstruction Loss** $\mathcal{L}_{recon}$**:** During each training iteration, we compute the pixel-wise reconstruction error using L1 loss $\mathcal{L}_{l1}$, while SSIM loss $\mathcal{L}_{ssim}$ is employed to assess the structural similarity between the predicted and ground truth images. Additionally, we incorporate LPIPS loss $\mathcal{L}_{lpips}$, leveraging a pre-trained VGG network as the backbone to evaluate perceptual similarity by extracting

Table S1: Loss functions applied during different optimization phases in the training process (§C).

| Loss | Warm-up | Gaussian opt | MLP opt | Backward opt | After Backward |
|---|---|---|---|---|---|
| $\mathcal{L}_{bone}$ | ✓ | | ✓ | ✓ | ✓ |
| $\mathcal{L}_{recon}$ | | ✓ | ✓ | ✓ | ✓ |
| $\mathcal{L}_{cycle}$ | | | | ✓ | |

high-level features. The overall reconstruction loss $\mathcal{L}_{recon}$ is then defined as:

$$\mathcal{L}_{recon} = \mathcal{L}_{l1} + \lambda_{ssim}\mathcal{L}_{ssim} + \lambda_{lpips}\mathcal{L}_{lpips}. \tag{10}$$

**Bone Regularization Loss $\mathcal{L}_{bone}$:** Given the absence of parametric templates for skinning regularization, we impose a rigid constraint during the skinning learning process to promote better convergence. Specifically, we sample $K = 1$ point at the midpoint of each bone. For leaf joints, we introduce a virtual joint located along the extension of the line connecting the joint and its parent, using this as the sample point. We then enforce that the skinning weights at these sampled points resemble one-hot vectors. The loss function is defined as:

$$\mathcal{L}_{bone} = \|\boldsymbol{W}_{sample} - \boldsymbol{W}_{gt}\|_2^2. \tag{11}$$

Here, $\boldsymbol{W}_{sample}$ represents the predicted skinning weights for the sampled points in the canonical space, and $\boldsymbol{W}_{gt}$ denotes the ground truth one-hot skinning weights.

**Cycle Consistency Loss $\mathcal{L}_{cycle}$:** Please refer to §3.4 in the main paper for details. The overall cycle consistency loss is then defined as:

$$\mathcal{L}_{cycle} = \lambda_{geo}\mathcal{L}_{geo} + \lambda_{skin}\mathcal{L}_{skin}. \tag{12}$$

We set the loss weights as follows: $\lambda_{ssim} = 0.01$, $\lambda_{lpips} = 0.5$, $\lambda_{bone} = 0.5$, $\lambda_{geo} = 1000$, $\lambda_{skin} = 10$ for all experiments. The application of these loss terms at different optimization phases is summarized in Table S1, which details the activation schedule of each loss function.

## C  IMPLEMENTATION DETAILS

**Training.** The training of TAGA is organized into several phases aimed at optimizing skinning and canonical appearance in a template-free environment. We begin with a warm-up phase to learn a rigid skinning field, during which the skinning weight field is optimized independently. Following this, we enter the main training phase. For the first 1.5K iterations, all components are frozen except for the 3D Gaussians. These Gaussians, driven by the pre-trained rigid skinning field, autonomously refine the positions and appearance in canonical space. Subsequently, we commence the optimization of the voxel-based skinning field, continuing to enforce the skinning weights regularization $\mathcal{L}_{bone}$. After 2.5K iterations, the backward mapping stage is activated, utilizing a cycle consistency loss $\mathcal{L}_{cycle}$ to address geometrical errors within the canonical space and further refine the skining field. The backward mapping phase is introduced only after the forward mapping reconstruction has stabilized, thereby ensuring that it serves to refine the geometry rather than disrupt it. To mitigate computational overhead, backward mapping is performed every 150 steps, with the positions and skinning weights of corrected Gaussians cached as soft constraints to continuously guide the optimization of canonical Gaussians. This strategy distributes the cost of ambiguity detection and correction across iterations, minimizing the impact on computational efficiency.

Due to the extremely limited pose in the PeopleSnapshot, some regions such as the armpits are occluded during training and remain unseen. To mitigate this issue and improve the model's ability to reconstruct these occluded areas, we add noise to the pose during the backward phase. Specifically, throughout the entire backward phase, we perturb the bone transfromation matrix $\boldsymbol{B}$ by adding noise sampled from a normal distribution $\mathcal{N}(0, 0.1)$ with a probability of $p = 0.5$.

Moreover, given the sparsity of skinning weights -— where each Gaussian is typically influenced by at most a few bones —we focus only on the bone with the highest posterior probability and its immediate neighboring bones when estimating the coarse skinning weights.

**Inference.** For Inference, we solely employ forward-mapping, leveraging the optimized skinning weights and the refined geometry. Similar to InstantAvatar [12], test-time pose refinement is also employed to enhance the results.

Table S2: Ablative experiments on voxel grid resolution for male-3-casual sequence of PeopleSnapshot (§E). The adopted hyperparameter is marked in red.

| Resolution | Memory | GPU | Novel pose | | |
|---|---|---|---|---|---|
| | | | PSNR↑ | SSIM↑ | LPIPS*↓ |
| $16 \times 16 \times 4$ | 4GB | 20min | 24.84 | 0.9475 | 34.9 |
| $32 \times 32 \times 8$ | 6GB | 24min | 27.89 | 0.9664 | 25.9 |
| $64 \times 64 \times 16$ | 10GB | 37min | **28.89** | **0.9685** | **23.1** |
| $128 \times 128 \times 16$ | 40GB | 70min | 28.22 | 0.9687 | 25.1 |

Table S3: Per-scene breakdown in novel view synthesis on ZJU-MoCap dataset (§C).

| Method | Subject 377 | | | Subject 386 | | | Subject 387 | | |
|---|---|---|---|---|---|---|---|---|---|
| | PSNR↑ | SSIM↑ | LPIPS*↓ | PSNR↑ | SSIM↑ | LPIPS*↓ | PSNR↑ | SSIM↑ | LPIPS*↓ |
| HumanNeRF [15] | 31.12 | 0.977 | 22.80 | 33.31 | 0.973 | 33.48 | 28.27 | 0.962 | 38.89 |
| NPC [16] | 31.80 | 0.974 | 16.31 | 33.01 | 0.965 | 30.69 | 27.26 | 0.948 | 42.85 |
| InstantAvatar [12] | 30.91 | 0.967 | 40.89 | 32.63 | 0.956 | 52.30 | 27.09 | 0.927 | 95.25 |
| TAVA [14] | 31.16 | 0.979 | 24.25 | 32.89 | 0.977 | 31.86 | 26.80 | 0.958 | 43.40 |
| TAGA (Ours) | **34.31** | **0.988** | 18.1 | **34.27** | **0.981** | **29.22** | **28.99** | **0.969** | 38.13 |

| Method | Subject 392 | | | Subject 393 | | | Subject 394 | | |
|---|---|---|---|---|---|---|---|---|---|
| | PSNR↑ | SSIM↑ | LPIPS*↓ | PSNR↑ | SSIM↑ | LPIPS*↓ | PSNR↑ | SSIM↑ | LPIPS*↓ |
| HumanNeRF [15] | 31.34 | 0.971 | 33.57 | 29.19 | 0.964 | 36.88 | 30.74 | 0.966 | 34.67 |
| NPC [16] | 32.31 | 0.963 | 29.76 | 29.08 | 0.953 | 35.69 | 31.14 | 0.957 | 29.74 |
| InstantAvatar [12] | 30.98 | 0.951 | 65.70 | 29.09 | 0.943 | 67.43 | 30.15 | 0.949 | 55.94 |
| TAVA [14] | 31.12 | 0.971 | 36.78 | 28.78 | 0.963 | 40.25 | 30.67 | 0.968 | 34.82 |
| TAGA (Ours) | **32.94** | **0.979** | **31.91** | **30.17** | **0.971** | **35.33** | **32.21** | **0.976** | **30.70** |

## D   IMPLEMENTATION DETAILS FOR BASELINES

**ZJU-Mocap [7].** For baseline methods InstantNVR [13], HumanNeRF [15], and GART [11], we utilize their official implementations and adopt the results reported in InstantNVR [13]. For InstantAvatar [12], we retrieve the ZJU-Mocap implementation from GauHuman and use the reported performance metrics [34]. For NPC [16], we obtain the official implementation for subject 387 in the ZJU-Mocap from the authors and apply the same parameter settings to evaluate other subjects within the dataset. For TAVA [14], which is not trained on the same data split as InstantNVR, we use its public code to retrain a new model.

**PeopleSnapshot [8].** For Anim-NeRF [65] and InstantAvatar [12], we utilize the reported results from InstantAvatar. For HumanNeRF [15], we retrain the model on the PeopleSnapshot dataset using the official code.

All reproduced baseline code and corresponding weights will be released to facilitate further research.

## E   ADDITIONAL DIAGNOSTIC EXPERIMENT

**Voxel Resolution.** Table S2 shows the impact of voxel grid resolution on novel pose performance. Generally, higher resolutions lead to higher accuracy but longer training time. A resolution of $64 \times 64 \times 16$ yields a good balance between accuracy and speed, achieving a PSNR of **28.89**, SSIM of **0.9685**, and LPIPS* of **23.1**, with a reasonable GPU memory usage of 10GB and a training time of 37 minutes. Lower resolutions, such as $16 \times 16 \times 4$, significantly degrade performance (with PSNR dropping to 24.84 and SSIM to 0.9475) while offering only marginal gains in speed. On the other hand, higher resolutions like $128 \times 128 \times 32$ require over an hour of training time and more than 4 times the memory usage, yet do not yield improvements in novel pose performance. This may be because the high resolution of the grid makes the voxel-based skinning field less stable.

## F   ADDITIONAL RESULTS

**Quantitative Results of Per-scene Breakdown on ZJU-Mocap.** We show the per-scene PSNR, SSIM and LPIPS on ZJU-MoCap in Table S3.

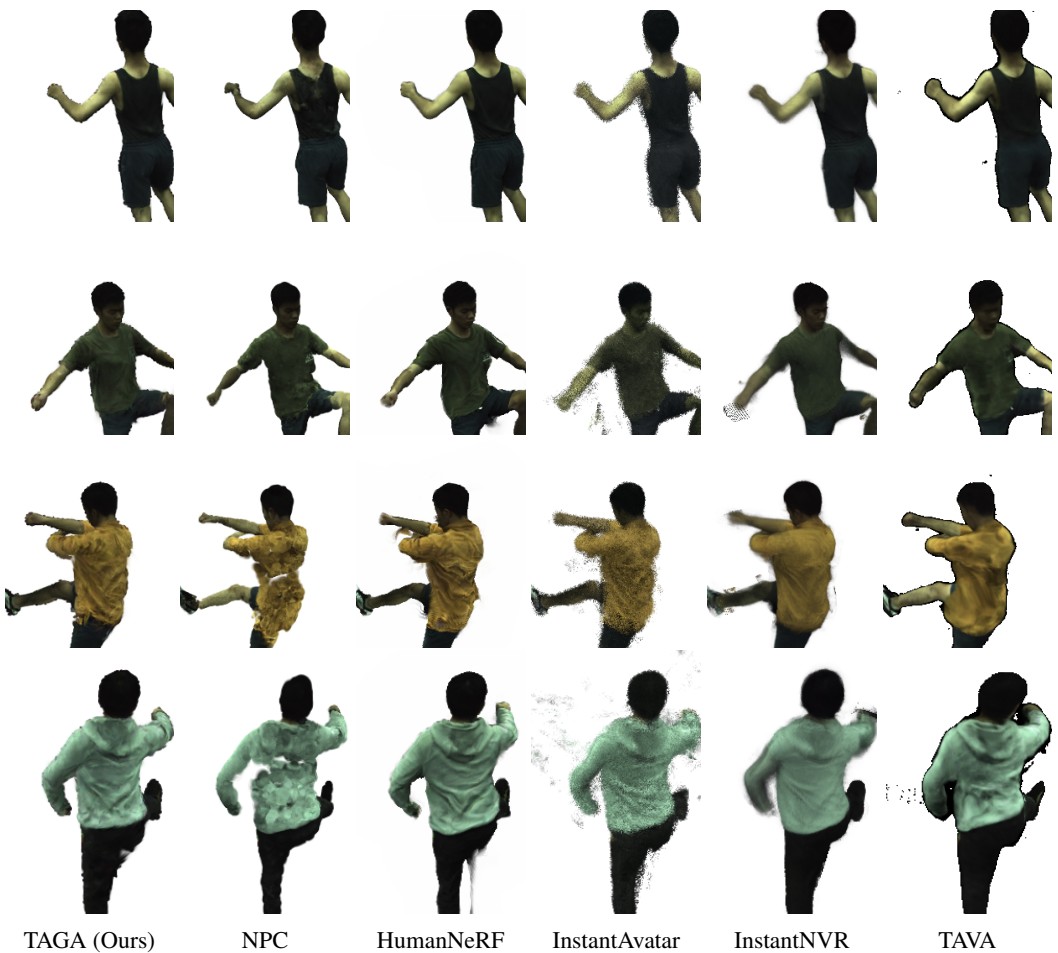

|                |            |            |               |            |         |
| TAGA (Ours)    | NPC        | HumanNeRF     | InstantAvatar | InstantNVR | TAVA    |

Figure S1: Qualitative comparison of novel view synthesis on zju-MoCap [7] (§F).

**Qualitative Results on ZJU-MoCap [7]**. In Fig. S1, we present novel view synthesis results for the remaining three subjects in the ZJU-MoCap dataset. NPC and InstantAvatar methods produce blurry reconstruction results, failing to capture fine details. HumanNeRF show relatively good visual quality, but some artifacts are noticeable around the edges. In contrast, TAGA achieves the best overall visual quality, effectively minimizing artifacts and preserving sharpness and detail throughout the entire image.

**Qualitative Results on PeopleSnapshot [8]**. In Fig. S2, we present additional novel view comparisons on the PeopleSnapshot dataset. HumanNeRF relies on pose-specific backward skinning to model canonical appearance. However, the limited variety of poses in the PeopleSnapshot hinders its performance, leading to incomplete reconstructions of the head and noticeable artifacts along the edges.

## G  DISCUSSION

**Limitation.** While TAGA demonstrates significant advancements in template-free modeling, it is important to acknowledge certain limitations that could impact its applicability in more complex scenarios: **i)** Non-rigid Deformations: TAGA struggles with excessively loose clothing or extreme non-rigid deformations. Such scenarios can disrupt the learning process for template-free skinning and pose challenges in generalizing to unseen poses. **ii)** Unseen Details and Artifacts: Although TAGA reduces the reliance on precise pose input and effectively addresses geometric artifacts of self-contact regions, it is still challenging to handle unseen details in the input data. Even when a Gaussian is placed correctly, issues such as holes or rendering artifacts may persist, especially in

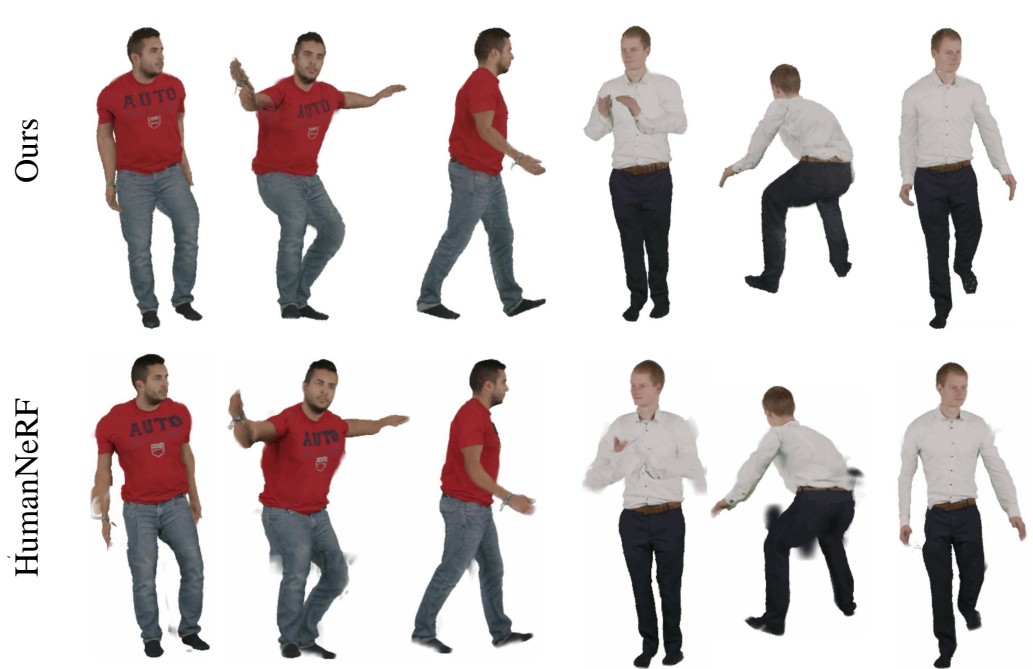

Figure S2: Additional qualitative results on PeopleSnapshot [8] (§E).

regions not visible in the input data. This limitation is a common challenge faced by other methods as well, indicating that further improvements are necessary.

**Future Work.** TAGA lays the groundwork for several promising future directions: **i)** While TAGA successfully reduces the dependency on parametric templates, it still relies on coarse pose or skeleton data. Future efforts could focus on integrating advanced skeleton extraction algoriithms or utilizing keypoints from existing models to better handle diverse object categories. **ii)** In this work, the anomaly detection algorithm we used is relatively basic. Future work could enhance this aspect by incorporating additional priors, such as category-specific classifiers, general image pretrained models, or even generative models. These improvements could help in identifying and correcting Ambiguous Gaussians, thereby addressing artifacts in avatar reconstruction. We believe that our proposed backward mapping strategy could become an attractive solution for 3D Gaussian representations to address underconstrained animatable avatar reconstruction scenarios.