# OpenReview forum: "TAGA: Self-supervised Learning for Template-free Animatable Gaussian Avatars"
_ICLR.cc/2025/Conference — ICLR 2025 Conference Withdrawn Submission_

### Official Review · Reviewer_a4ss · 2024-10-23

**Soundness:** 2
**Presentation:** 3
**Contribution:** 3
**Rating:** 5
**Confidence:** 5

**Summary:**

The paper marries Gaussian splatting with template-free human rendering. To resolve the amplified ambiguities in the template-free setting, it refines the learned blend skinning weights in the observation space using GMM priors and EM. The corrected blend skinning weights are then employed to supervise the Gaussians and blend skinning weights in the canonical space. Extensive results show that the proposed method surpasses the state-of-the-art template-free approaches in terms of rendering quality as well as training and rendering speed.

**Strengths:**

* The paper is well-written and easy to follow. All proposed components are well-motivated and described in detail.
* It demonstrates the effectiveness of the proposed approach with comprehensive experiments. It outperforms the template-free baselines both quantitatively and qualitatively.

**Weaknesses:**

* The benefits of template-free methods are unclear to me due to the following reasons: 1) The authors claimed that template-based methods “require labor-intensive 3D scanning and manual annotation”. However, current human mesh recovery can efficiently and accurately predict SMPL parameters. Template-based methods such as GART and GoMAvatar showcase in-the-wild videos where the human poses and shapes are predicted from off-the-shelf tools such as ReFit [1]. Therefore the templates are not too expensive to acquire. 2) Although the proposed method doesn’t use any priors from human templates, it still requires heavy handcrafted priors to constrain the solutions, for example, the GMM model. 3) Both the training speed and rendering quality of template-free methods still lag behind those of template-based methods. I would appreciate it if the authors could clarify how the manually designed priors in this work outperform the priors derived from human templates. Do the proposed priors offer better generalization to certain scenarios? How does TAGA's performance compare to template-based approaches with predicted SMPL parameters as inputs?

* The method heavily relies on the anomaly Gaussian detection and refinement with GMM priors and EM. The effectiveness of the EM process is not shown in the paper. Could the authors provide an ablation study or visualization showing how the EM process improves the anomaly Gaussian detection and refinement over iterations?

* It is unclear how the method could generalize to in-the-wild scenes where subject masks and poses are predicted and therefore less accurate. Also, the poses in in-the-wild scenes can be more challenging compared to ZJU-MoCap and PeopleSnapshot.

[1] Wang, Yufu, and Kostas Daniilidis. "Refit: Recurrent fitting network for 3d human recovery." Proceedings of the IEEE/CVF International Conference on Computer Vision. 2023.

**Questions:**

In addition to the concerns listed in weaknesses,
* Is the prior still effective when the body parts are close in the monocular video, e.g. in UBC-Fashion, subjects’ arms sticking to the torso?
* Is TAGA able to learn the correct blend skinning weights for challenging clothes, e.g., dresses?

---

### Official Review · Reviewer_7GHD · 2024-10-29

**Soundness:** 3
**Presentation:** 3
**Contribution:** 3
**Rating:** 5
**Confidence:** 4

**Summary:**

This paper introduces a template-free, Gaussian-based method to reconstruct animatable avatars from monocular video input. The proposed method has two core designs to achieve superior visual quality and fast training and real-time rendering. The first core design is a self-supervised method for guiding both geometry and skinning learning. Reconstructing consistent template-free avatars is challenging due to lack of guidance such as predefined shapes or skinning anchors. To address this, TAGA leverages the one-to-one correspondence between canonical and observation spaces. The second core design is the new backward-mapping strategy that integrates anomaly detection to alleviate ambiguous Gaussians that may lead to artifacts. Extensive experiments demonstrate superior visual quality over previous template-free methods while achieving much faster training and rendering.

**Strengths:**

1.	The presentation of the paper is clear and easy to follow.
2.	The proposed method achieves significant improvements on the training speed and rendering speed by utilizing the Gaussian representation.
3.	The voxel-based skinning field for forward deformation achieves fairly good and robust results.
4.	The backward-mapping strategy that utilizes anomaly detection can alleviate unrealistic geometric artifacts and demonstrate better visual quality than previous template-free methods.

**Weaknesses:**

From the perspective of human avatar reconstruction, SMPL tends to be a strong prior for high-quality reconstruction. On the other hand, template-free methods are often used to reconstruct complex human avatars like humans with dress or animals (TAVA showcases such results). However, while the method claims template-free as one of the main contributions, the paper does not demonstrate such results. If the scope of experiments lies in the reconstruction of animatable human avatars, then Gaussian-based methods like GART (SMPL-based) has better results regarding visual quality (can be seen in Table.2) and comparable efficiency. This needs to be addressed by the authors in the discussion section. Comparing GART and TAGA on complex avatars with loose clothing or animal subjects can better demonstrate the advantage of template-free methods.

**Questions:**

The reconstruction results of TAGA in Figure.5 have black pixels around the avatars, TAVA also appears to have such results while other methods don’t. This can lead to worse visual quality. Is it because of the mask or such methods tend to generate such results? The cause of the black pixels and the potential impact need to be addressed.

---

### Official Review · Reviewer_K6hj · 2024-11-04

**Soundness:** 2
**Presentation:** 2
**Contribution:** 2
**Rating:** 5
**Confidence:** 3

**Summary:**

This work introduces an approach for creating animatable avatars from monocular videos. The key components of the method include: relying on Gaussians attached-to-bones as the rendering representation, voxelized skinning field (which according to the authors provides better generalization over more widely used MLPs), GMM-based technique to fix regions with ambiguous mapping, as well as a backward mapping strategy utilizing the ambiguous gaussian detection. Experimental results on ZJU-Mocap and PeopleSnapshot suggest that method performs better than several other template-free baselines in terms of quality and speed.

**Strengths:**

+ Provides a way to learn an animatable avatar without the need of existing mesh templates - which can help simplify overall pipelines.
+ Method seems to lead to fewer artifacts compared to other template-free methods.
+ Quantitative results suggest improved performance over other template-free methods.
+ The training speed is significantly higher than some of the existing methods (although it is probably mostly due to reliance on gaussians as primitives).

**Weaknesses:**

- A lot of the claimed improvements - such as speed of training / quality - could actually be largely due to the use of a different representation - gaussian splats - which probably cannot be considered a novel contribution at this point.
- The method is person-specific, meaning that it requires training per individual, and still requires 30 minutes to train on a video.
- The GMM-based technique for fixing "ambiguous guassians" seems to be very ad-hoc and is not used in a joint optimization with the model parameters.
- The overall quality is limited - and to judge it fully ideally one needs to provide comparison with ground truth in images, not just competing methods - to understand how well identity is preserved.
- In some cases it is hard to actually reason about the quality of the method, given the poor quality of the datasets used (in particular ZJU-Mocap). I am not sure if visuals on that dataset are very informative. Also, in that dataset person is rotating 360 views, thus the claim about method being monocular is somewhat weaker.
- Potential missing comparisons: Animatable Gaussians (CVPR 2024), ExAvatar (ECCV 2024).
- (Minor) Although authors suggest that being template-free is important - why is it actually important - is not clearly explained in the paper.

**Questions:**

- Would it be possible to provide comparisons to ground truth in the visuals?
- How does the method stack against recent methods such as Animatable Gaussians and ExAvatar?
- (Minor) Providing a more coherent explanation of why template-free is a critical design choice would be helpful.

---

### Official Review · Reviewer_jyYf · 2024-11-05

**Soundness:** 2
**Presentation:** 2
**Contribution:** 2
**Rating:** 5
**Confidence:** 4

**Summary:**

This paper proposes a template-free Gaussian avatar. As far as I know, this is the first template-free Gaussian human avatar. The experiment results demonstrate improved performance over those template-free NeRF methods. This is reasonable. However, the paper lacks a deep insight description of the technical contribution. From my point of view, the advantage of template-free property is quite similar with those NeRF-based template-free methods. It would be good to make more clear about why the method can acchieve fast training and high quality rendering, even using a template-based method. From my point of view, template-free alone is not good enough and fast training and high quality rendering are more attractive to me. Also, quantitative experimental results shown in the paper do not validate the improved performance over methods like HumanNeRF. The paper does not provides results on high quality multiview datasets and especially loose cloth human or animals, lacking the validation of template-free benefit.

**Strengths:**

This paper presents a novel approach for constructing template-free, animatable human avatars using 3DGS. To address the skinning ambiguities between adjacent body parts, the authors propose to regularize the canonical Gaussian points through a bone-based Gaussian mixture model and an EM optimization algorithm. By treating these Gaussian points as deformation anchors, the canonical geometry can be further refined in a self-supervised manner. The experimental results demonstrate better performance compared to previous template-free methods, both qualitatively and quantitatively.

1.  The paper proposes the first template-free Gaussian human avatar.
2. Technical details are reasonable.
3. Paper writing is most clear. But insight is not that clear.
4. Results are mostly reasonable.

**Weaknesses:**

1. Lack of video demo to show the performance on dynamic appearance and dynamic details. This is important to see if the results suffer temporal jitter or not. Also, this is important for qualitative evaluation, as figure results can be delibrately choosed from the results.
2. The results shown in this paper are all tight cloth humans, with loose cloth humans and animals. This could not demonstrate the advantage of template-free method. Is the proposed method fit for loose cloth humans and animals?
3. The main improvement of this paper is the training speed and the rendering quality, quantitatively compared with other methods like NPC, HumanNeRF, TAVA, as shown in figure 2. However, this performance gain seems to come most from the using of Gaussian representation. Methods using templates and Gaussian splatting have emerged these days. It is better to compared with these method to see the performance differences on training time and rendering quality，especially since the results shown in this paper are tight clothed humans.
4. Based on 3), I would think if using template and Gaussian splatting, the training time would be further lower, and the rendering quality would be further improved. Is this the truth? Can you discuss the trade-offs between template-free and template-based approaches when combined with Gaussian splatting, specifically in terms of training time and rendering quality.
5. Qualitative results shown in Fig.5 show that, HumanNeRF is much than the proposed method, as the face and the clothes are much more clear than the proposed method.  Can you please address this apparent contradiction and provide more detailed analysis of where their method outperforms HumanNeRF quantitatively despite the qualitative differences.
6. I would suggest using high quality multiview human performance sequences like ActorShq or Thuman4.0 in the experiment sessions. It would be more clear to see if the results are good or not. I know that the two datasets you used in the paper are widely used in monocular human avatar, but I would still want to know more about the reason of choosing these datasets. Also, is it possible for you to provide your results on ActorShq dataset?

**Questions:**

See the weaknesses.

---

### Official Review · Reviewer_e1ch · 2024-11-05

**Soundness:** 3
**Presentation:** 3
**Contribution:** 2
**Rating:** 5
**Confidence:** 4

**Summary:**

This paper presents a method for learning animatable 3D Gaussian avatars from monocular videos. Unlike existing works, the proposed method does not rely on an explicit 3D mesh template. Instead, the Gaussian positions are initialize from a Gaussian distribution around each bone, and the skinning field is initialized and stored in a low-resolution voxel grid. During training, an ambiguous Gaussian correction strategy is introduced to ensure all the Gaussian points have plausible skinning weights and canonical positions. Experiments show that the proposed method is able to reconstruct plausible avatars from monocular videos without any template inputs.

**Strengths:**

* The proposed method can reconstruct an animatable Gaussian avatar without the need of mesh templates.

* The authors propose a strategy to detect "ambiguous Gaussians" that may have unreasonable positions or skinning weights. With such a detection strategy, these ambiguous points can be corrected, leading to a more plausible shape reconstruction.

**Weaknesses:**

* The authors did not provide any video results, which makes it difficult to evaluate the animation quality. Providing video results is a common practice in the research field of avatar modeling technology.

* The authors only conducted experiments on human bodies. As a template-free method, it can be applied for other creatures, eg., pigs and dogs. Previous template-free methods like TAVA demonstrated their ability in modeling different creatures in their paper, so I encourage the authors to conduct a similar experiments to better showcase the capability of the proposed method.

*  In Abstract, the authors claims a really impressive speedup ("60x faster in training" and "560x faster in rendering"). However, this advantage is mainly brought by the Gaussian splatting itself, rather than the technical contributions of this paper. Additionally, given that many existing works have already applied Gaussian splatting in the task of avatar modeling, I think the authors should tone down this advantage.

**Questions:**

See [weaknesses].

---

### Note · Authors · 2024-11-13

**Comment:**

We would like to formally withdraw our paper from the ICLR 2025 submission process. After thorough consideration, we have decided to revise and improve our work based on new findings and constructive feedback. We believe these revisions will lead to a stronger and more impactful contribution. We are grateful for the time and thoughtful feedback provided by the reviewers and Area Chairs.

**Withdrawal Confirmation:**

I have read and agree with the venue's withdrawal policy on behalf of myself and my co-authors.